# Modified Atkins diet induces subacute selective ragged-red-fiber lysis in mitochondrial myopathy patients

Sofia Ahola[1], Mari Auranen[1,2], Pirjo Isohanni[1], Satu Niemisalo[3], Niina Urho[2], Jana Buzkova[1], Vidya Velagapudi[4], Nina Lundbom[5], Antti Hakkarainen[5], Tiina Muurinen[6], Päivi Piirilä[6], Kirsi H Pietiläinen[2,7] & Anu Suomalainen[1,2,8,*]

## Abstract

Mitochondrial myopathy (MM) with progressive external ophthalmoplegia (PEO) is a common manifestation of mitochondrial disease in adulthood, for which there is no curative therapy. In mice with MM, ketogenic diet significantly delayed progression of the disease. We asked in this pilot study what effects high-fat, low-carbohydrate "modified Atkins" diet (mAD) had for PEO/MM patients and control subjects and followed up the effects by clinical, morphological, transcriptomic, and metabolomic analyses. All of our five patients, irrespective of genotype, showed a subacute response after 1.5–2 weeks of diet, with progressive muscle pain and leakage of muscle enzymes, leading to premature discontinuation of the diet. Analysis of muscle ultrastructure revealed selective fiber damage, especially in the ragged-red-fibers (RRFs), a MM hallmark. Two years of follow-up showed improvement of muscle strength, suggesting activation of muscle regeneration. Our results indicate that (i) nutrition can modify mitochondrial disease progression, (ii) dietary counseling should be part of MM care, (iii) short mAD is a tool to induce targeted RRF lysis, and (iv) mAD, a common weight-loss method, may induce muscle damage in a population subgroup.

**Keywords** mitochondrial myopathy; modified Atkins diet; PEO; ragged-red-fibers

**Subject Categories** Genetics, Gene Therapy & Genetic Disease; Metabolism

See also: **RDS Pitceathly & C Viscomi** (November 2016)

## Introduction

Mitochondrial disorders are among the most common inherited metabolic diseases affecting approximately 1 in 2,000–5,000 individuals (Skladal *et al*, 2003; Thorburn, 2004). These disorders are most often caused by dysfunction of the oxidative phosphorylation (OXPHOS) system, which comprises the enzymes of the respiratory chain (RC) and the ATP synthase embedded in the inner mitochondrial membrane. These molecular micromachines transform the nutrient energy to ATP, the common energy currency of cells. RC deficiency can be caused by mutations in nuclear genes, encoding proteins involved in RC synthesis, assembly, structure, or maintenance, or by primary mutations of mitochondrial DNA (mtDNA). Progressive external ophthalmoplegia (PEO) is a common manifestation of adult-onset mitochondrial myopathy (MM), often caused by sporadic heteroplasmic single deletions, point mutations of mtDNA, or multiple mtDNA deletions caused secondarily by mutation of nuclear genes involved in mtDNA maintenance (Ylikallio & Suomalainen, 2012). MM manifests typically in the early adulthood, with progressive ptosis, ophthalmoplegia, and exercise intolerance (Zeviani *et al*, 1989). Many patients benefit from palliative care, including surgical blepharoplasty, but no curative treatments exist.

Low-carbohydrate ketogenic diet has been utilized to treat refractory epilepsies in children and has also been shown to reduce seizures in individuals with mitochondrial disease (Kang *et al*, 2007; Joshi *et al*, 2009). It also has been evaluated as a therapy for metabolic syndrome and type 2 diabetes, while a form of ketogenic diet, Atkins diet, is used widely for weight loss (for a review, see Paoli *et al*, 2013). The diet reduces blood glucose and improves insulin responses and lipid profiles in healthy adults (Westman *et al*, 2002) as well as in diabetic individuals (Gumbiner *et al*, 1996; Boden *et al*, 2005) and has been considered relatively safe.

1   Research Program of Molecular Neurology, Biomedicum Helsinki, University of Helsinki, Helsinki, Finland
2   Clinical Neurosciences, Neurology, University of Helsinki and Helsinki University Hospital, Helsinki, Finland
3   Obesity Research Unit, Research Programs Unit, Diabetes and Obesity, University of Helsinki, Helsinki, Finland
4   Metabolomics Unit, Institute for Molecular Medicine Finland FIMM, University of Helsinki, Helsinki, Finland
5   Department of Radiology, University of Helsinki and HUS Radiology, Helsinki Medical Imaging Center, Helsinki, Finland
6   Department of Clinical Physiology and Nuclear Medicine, Laboratory of Clinical Physiology, Helsinki University Hospitals, Helsinki, Finland
7   Department of Medicine, Division of Endocrinology, Helsinki University Central Hospital, Helsinki, Finland
8   Neuroscience Center, University of Helsinki, Helsinki, Finland
    *Corresponding author. Tel: +358 9 47171965; E-mail: anu.wartiovaara@helsinki.fi

We have previously shown that in MM mice, ketogenic diet can slow down progression of the disease and improve metabolic and lipidomic parameters (Ahola-Erkkila *et al*, 2010). Encouraged by the beneficial effects in mice, we performed a pilot study in PEO/MM patients, as well as for age- and gender-matched voluntary controls with ketogenic "modified Atkins diet" (mAD). We demonstrate that mAD can dramatically modify the disease course of adult-onset MM by inducing muscle damage, especially targeting respiratory chain-deficient fibers. The patients recovered well and showed a moderate long-term increase in muscle strength.

## Results

### Study subjects and the diets

Table 1 summarizes the study subjects and Fig 1A the study design. All of the five patients had PEO, mild proximal muscle weakness, and exercise intolerance. For every patient, two gender- and age-matched, healthy voluntary controls were recruited (males, *n* = 6; females, *n* = 4; age range: 35–61 years). Fig 1A summarizes the study protocol. Normalized isocaloric standard diet (ND) was introduced to all study subjects for 2 weeks, after which they were gradually switched to mAD for 4 weeks. The pre-diet carbohydrate intake of PEO patients was 41–48% (control mean = 43%), 14–20% (18) proteins and 27–38% (34) fats of the total calorie intake. During mAD, study subjects increased their protein and fat intake and reduced gradually carbohydrate intake to 3–9% (Table EV1 and Fig EV1A). The intake of saturated, monounsaturated, and polyunsaturated fats all increased, but especially the two latter (Table EV1).

### Modified Atkins diet induced acute muscle damage in all PEO patients

The clinical status of the subjects was stable during their 2 weeks on ND and the week of gradual transition to mAD, as documented by clinical follow-up and the responses to questionnaires. The plasma β-hydroxybutyrate levels increased in all the study subjects, indicating ketosis and good compliance to the study (Fig 1B). However, after 3–4 days of full mAD all of the PEO patients experienced similar, gradually progressing muscle symptoms: burning sensation and pain in their leg muscles which was enhanced by exercise. These symptoms ascended to the lower back muscles, and

then to the arms and neck. Patients also reported headache and tiredness. At the time, serum analysis of all the patients showed a progressive increase of serum CK, alanine aminotransferase, and myoglobin values (Figs 1C and D, and EV1C). Plasma FGF21 concentrations did not significantly change after mAD, even after 2.5 years (Fig EV1B). Kidney function and serum creatinine concentrations remained stable. Because of progressive symptoms in all of the five PEO patients, the trial was prematurely discontinued, with the maximum duration being 11 days on mAD. The laboratory tests and muscle biopsy samples were collected just before diet termination. The plasma CK and myoglobin concentrations decreased to pre-diet levels within 2 weeks. The diet-associated muscle symptoms gradually attenuated, and 1 month after mAD, the subjective and objective clinical status of the patients was comparable to the pre-diet state.

All of the control subjects tolerated mAD well and had no muscle symptoms or elevated plasma CK values, and they all continued the diet for the full 4 weeks. However, their sampling scheme, including the second muscle biopsy, was timed similar to their corresponding patients. All patients and controls showed an increase in plasma urea, indicating amino acid oxidation, and a slight reduction in plasma triglyceride values, but cholesterol, glucose, and insulin levels remained normal (Fig EV1D–H). These results indicate that mAD causes subacute muscle damage in MM patients, independent of the underlying genetic defect, but has no such effects for healthy muscle.

### Muscle fiber degeneration in PEO, especially in ragged-red fibers

Prior to mAD, the *vastus lateralis* muscle of PEO patients showed typical MM findings in their ultrastructure, with enlarged mitochondria containing distorted or concentric cristae as well as paracrystalline and electron-dense inclusions (Fig 1E–G). After mAD, most muscle fibers of PEO patients still had normal morphology, but some fibers showed acute degenerative changes, with central vacuoles containing cell debris surrounded by autophagosome membrane-like structures (Fig 1H–K). A total of 0–4.5% of muscle fibers were necrotic, with infiltrated macrophages (Fig 1I and K). These individual dying muscle fibers often showed clear subsarcolemmal mitochondria with paracrystalline inclusions (Fig 1J), typical for RRFs. The short mAD showed no significant effects on the overall histochemical COX activity, mtDNA copy number, or mtDNA deletions in the patients' muscle (Fig EV1I–K). These results

**Table 1.  Clinical and molecular characteristics of PEO patients.**

| Subjects | Gender | Age | Age of onset | Disease gene | Main clinical symptoms | COX-negative muscle fibers (%) | RRFs (%) | Duration of mAD |
|---|---|---|---|---|---|---|---|---|
| 1 | F | 36 | 22 | single mtDNA del | ptosis, PEO, exercise intolerance | 61 | 7 | 8 days |
| 2 | M | 54 | 20 | multiple mtDNA del[a] | ptosis, PEO, exercise intolerance | 11 | 4 | 9 days |
| 3 | M | 52 | 30 | multiple mtDNA del[a] | ptosis, PEO, exercise intolerance | 13 | 5 | 8 days |
| 4 | M | 40 | 31 | multiple mtDNA del[a] | ptosis, PEO, exercise intolerance | 22 | 4 | 11 days |
| 5 | F | 62 | 30 | single mtDNA del | ptosis, PEO, exercise intolerance | 27 | 3 | 4 days |
| Ctrls | F = 4 M = 8 | 35–61 | – | – | – | – | – | 4 weeks |

RRF, ragged-red fibers; F, female; M, male; del, deletions; mAD, modified Atkins diet.
[a]Twinkle duplication mutation of p.352–364.

                                                                              

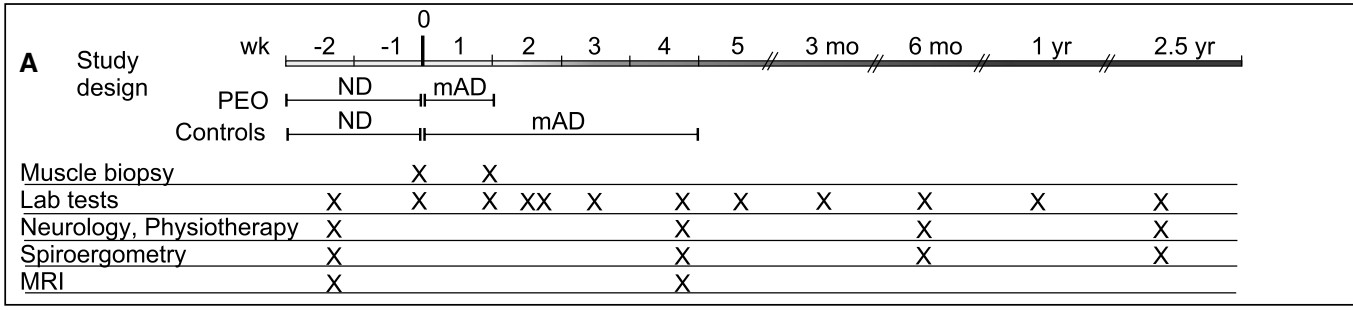

Figure 1.

**Figure 1.   Modified Atkins diet induces ragged-red-fiber lysis in patients with mitochondrial myopathy.**

A      Modified Atkins diet (mAD) study protocol. PEO patients followed mAD for 4–11 days; the healthy control subjects for 4 weeks. Sampling times of controls were matched with their corresponding PEO patients.

B–D   Plasma beta-hydroxybutyrate (ketone body; B), creatine kinase (CK; C), and myoglobin (D) concentrations. Dashed line: upper limit of control range. The individual patients are shown separately, and controls as mean with standard deviation (*n* = 10). Arrow: mAD endpoint in PEO patients; lines under graphs: duration of mAD.

E–G   PEO patient (P1) muscle before diet: (E) Ultrathin section of abnormal muscle fiber (arrowhead) with subsarcolemmal accumulations of mitochondria. Scale bar, 50 μm. (F) Electron micrograph of subsarcolemmal accumulation of abnormal mitochondria (same cell as in E). (G) Enlargement of the area marked in (F); mitochondria with paracrystalline "parking lot" inclusions (arrow) and concentric cristae (asterisk). Scale bar, 1 μm.

H–J   PEO patient (P1) muscle after mAD (H–J): (H) ultrathin section, lytic muscle fiber with centrally located organellar debris (arrow). Scale bar, 50 μm. (I) Electron micrograph of a degrading muscle fiber with an invaded macrophage (arrow) (same cell as in H). (J) Enlargement of the fiber in (I): paracrystalline inclusions within mitochondria (asterisk). Scale bar, 1 μm.

K      Quantification of necrotic or apoptotic cells in PEO muscle from ultrathin sections. n.a., not available.

explain the increased CK and myoglobin values and the muscle pain of the patients and indicate that mAD acutely and selectively damages RRFs, the muscle fibers with most prominent RC deficiency.

## Increased caspase-3 activation and stalled autophagy in PEO muscle

As muscle ultrastructure showed muscle fiber degeneration, we immunostained the muscle sections for the proapoptotic cleaved caspase-3, together with mitochondrial marker, to mark the affected fibers with mitochondrial proliferation. Occasional muscle fibers showing caspase-3 activation were detected in both pre- and post-mAD in PEO, and the proportion of these fibers did not depend on the diet, suggesting that caspase-3 activation and potential apoptosis did not contribute to degenerative changes—indeed, the necrotic fibers did not show increased amount of caspase-3 (Fig 2A and B). These results indicated that mAD induced fiber degeneration in PEO patients mostly by necrosis.

As some fibers showed central accumulations of autophago-some-like structures in EM, we studied the amounts of p62, a receptor for ubiquitinated cargo destined for autophagy (Bjorkoy *et al*, 2005). Western blot analysis indicated increased amount of p62 (two-tailed Mann–Whitney test, *P* = 0.0121; Fig 2C), suggesting that the disease itself resulted in reduced autophagic flux, leading to decreased receptor clearance. These results were corroborated by immunohistochemical detection of p62, showing p62 positivity especially in affected fibers with high mitochondrial content (Fig 2D–G). However, the short mAD did not significantly

change p62 protein amounts in PEO patients' muscle. To study further the autophagy pathway, we analyzed the presence of an autophagosome membrane component, microtubule-associated protein 1 light chain 3B (LC3). This protein showed colocalization with p62 in fibers with high mitochondrial content (Fig 2H) and a mosaic-staining pattern in patient muscles compared to a uniform pattern in healthy controls (Fig 2I). Only after mAD, occasional fibers showed positive accumulation of LC3 (Fig 2I). To clarify whether mAD-induced muscle damage led to satellite cell activation, we analyzed pax-7 protein amount (Hawke & Garry, 2001). ND PEO muscle samples showed increased amounts of pax-7 (two-tailed Mann–Whitney test, *P* = 0.006), suggesting basal satellite cell activation in MM, but the activation was not further enhanced by the short mAD in the acute stage of muscle damage (Fig 2C). These results suggest that mAD did not increase autophagic flux or regenerative mechanisms in the acute myolytic stage. Control individuals showed no changes in any of the markers.

## Mild improvement in muscle strength in PEO patients in 2.5-year follow-up

We asked whether the mAD-induced RRF lysis in our patients would affect long-term muscle function, hypothesizing that the damage might induce satellite cell fusion and regeneration. In 2.5 years of follow-up, three out of four patients showed slight improvement in muscle strength of leg and back muscles and in six-minute walking distance (Figs 3A–C and EV2A–C). All of the patients had improved lower extremity extension strength, and three out of four had

**Figure 2.   Mitochondrial myopathy patients show caspase-3 cleavage, satellite cell activation, and stalled autophagy, not affected by modified Atkins diet.**      ▶

A, B   Cleaved caspase-3-positive muscle fibers (*). PEO patient (P3) on normal diet (A; inset: caspase staining in healthy control individual) and after modified Atkins diet (B). Arrowhead: caspase-negative necrotic fiber.

C      Western blot analysis of autophagy receptor p62 and pax7 marking muscle satellite cell activation; left, blot; right, signal quantification in each patient.

D, E   Immunohistochemical analysis of p62 and mitochondrial mass marker in consecutive sections. Asterisks indicate p62-positive fibers and corresponding affected ragged-red fibers.

F, G   Immunohistochemical analysis of p62 and mitochondrial mass marker after mAD; consecutive sections. Arrowheads indicate fibers that show p62 positivity but have normal mitochondrial mass.

H      p62- and LC3-positive ragged-red fibers (*) in PEO patient (P2) on normal diet.

I      Muscle fibers showing granular LC3-positivity in PEO patient (P1) after mAD. Arrowhead indicates fiber that is enlarged in the inset figure.

Data information: Immunohistochemistry was performed on frozen sections. Scale bars: 50 μm (A–G, I) and 100 μm (H). Abbreviations: PEO, progressive external ophthalmoplegia; ctr, healthy control individual; ND, normal diet; mAD, modified Atkins diet; Tom20, mitochondrial outer membrane transporter 20; p62, autophagy receptor protein 62; pax7, paired box protein 7.

Source data are available online for this figure.

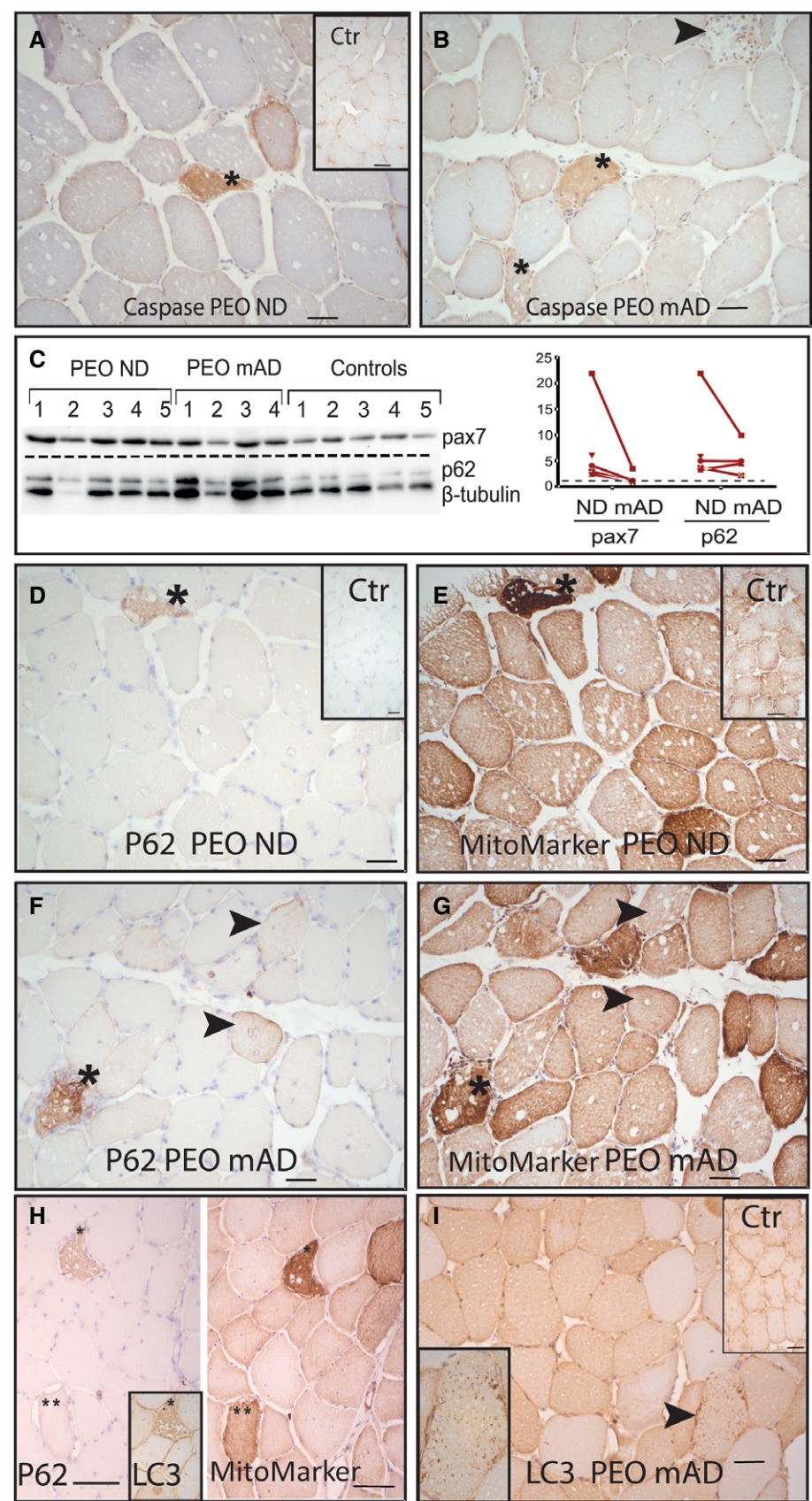

**Figure 2.**

improved static back muscle strength. The youngest patients improved in all muscle strength tests. These results implied subtle performance improvement, without any remarkable reported changes in patients' daily physical exercise, suggesting that the fiber degeneration favoring RRFs may have affected the amount of sick fibers and/or promoted regeneration with subtle long-term beneficial consequences for muscle strength.

## PEO patients are glycolytic and unable to switch to beta-oxidation

The findings from the patients' serum tests and muscle samples suggested that upon low dietary carbohydrate content, RRFs were unable to utilize ketones and fatty acids as their fuel and were highly dependent on glucose as a carbon source. Accordingly, light

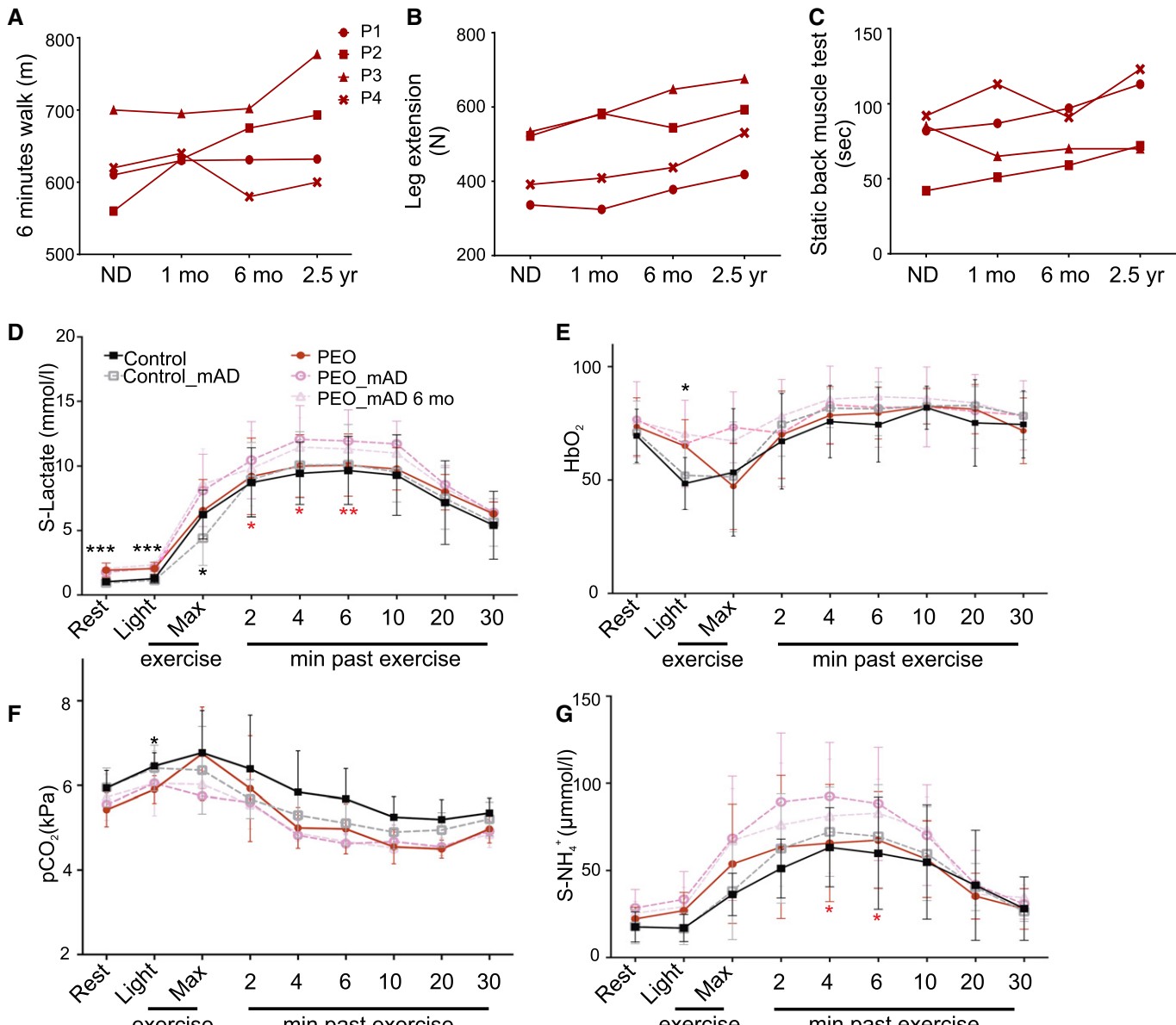

**Figure 3.  Physiological and performance test results after mAD.**

A–C   Muscle strength and performance of PEO patients (P1-P4) on normal diet (ND) and after modified Atkins diet (mAD) (1 and 6 months and 2.5 years after initiation of diet).

D–G   Spiroergometry results from PEO patients and control subjects on normal diet (ND), as well as 1 and 6 months after mAD initiation. PEO patients had higher serum lactate levels, and attenuated respiratory response to exercise. These effects were further enhanced after mAD and remained after 6 months of the study. mAD increased patients' serum ammonium ($NH_4^+$) levels. Data are presented as mean ± SD.

Data information: Abbreviations: Max, maximal exercise level; $HbO_2$, oxidized hemoglobin; $pCO_2$, partial pressure of $CO_2$. Statistical analyses: Student's *t*-test (two-tailed). Asterisk above the data point: significant difference between PEO and controls. Asterisk below the data point: significant difference between ND and mAD within a group (color-coded). *$P < 0.05$, **$P < 0.01$, ***$P < 0.001$.

exercise stimulated a lactate increase in PEO patients in spiroergometry test, which was further enhanced post-mAD (Fig 3D). The blood ammonium values during ND were similar in PEO patients and in controls, indicating equal usage of proteins as a fuel (Fig 3G). Ammonium production in exercise remained slightly increased in 6 months of follow-up of PEO patients. They also showed slightly decreased oxygen extraction from the blood (Fig 3E) and lower $CO_2$ production (Fig 3F) after mAD, which was sustained in 6 months of follow-up, suggesting that the switch to oxidative metabolism in exercise was compromised. MRI and MRS analyses of PEO patients showed increased liver and visceral adipose tissue (VAT) fat levels as well as elevated intramyocellular lipid (IMCL) and extramyocellular lipid (EMCL) levels in the skeletal muscle. PEO patients also had less double-bond lipids in their subcutaneous adipose tissue ($CH_2/CH_3$). The short mAD did not, however, significantly affect these parameters (Fig EV2D). mAD decreased slightly the amount of subcutaneous adipose tissue (SAT) in both patients and controls (ANOVA, $P < 0.001$) with slight weight loss (PEO: 1–4 kg; and controls: 1–2 kg) despite the isocaloric diet design. These results indicate that PEO is associated with accumulation of fats in the liver, adipose tissue, and muscle and that PEO patients have a glycolytic metabolic state.

### Transcriptome consequences of mAD: suppression of mitochondrial oxidative metabolism in PEO muscle, and induction in healthy muscle

Global muscle transcriptomic analysis, comparing pre- to post-mAD expression patterns in each patient, indicated widespread, highly significant down-regulation of pathways linked to carbohydrate-driven oxidative metabolism in PEO muscle. These changes spanned from pyruvate dehydrogenase and acetyl-CoA synthesis to tricarboxylic acid cycle and oxidative phosphorylation (Fig 4A and B). Consistent with the lytic changes, extracellular pathways associated with degeneration and inflammation were induced. In controls, however, mAD activated oxidative metabolism, up-regulating pathways of mitochondrial fatty acid transport and β-oxidation, protein oxidation, and ketogenesis (Fig 4A and B). These pathways were not induced in PEO, indicating that the transcriptional response inducing lipid and ketone body utilization was lacking in MM muscle. Our results also indicate that in controls, mAD strongly induced a fasting-like response of oxidative metabolism.

### mAD modifies plasma metabolome

Our quantitative targeted metabolomics analysis of 88 metabolites of plasma indicated that PEO patients responded at the metabolome level quite similarly to the controls (Figs 5A–D and EV3A and B). However, mAD normalized some disease-associated metabolite changes in PEO patient plasma, such as asparagine and alanine, the latter being a known biomarker for mitochondrial disease (Fig 5A) and xanthine (Fig 5D). In both patients and controls, mAD increased significantly the levels of branched chain amino acids (BCAAs) valine, isoleucine, and leucine (Fig 5A and B). Creatine amount was increased in PEO patients' plasma and further increased after mAD both in PEO patients and in controls, which together with decreased precursors (glycine, arginine, and guanidinoacetic acid) and increased creatine:creatinine ratio suggested

increased creatine synthesis, thereby increasing availability of high-energy phosphates buffering ATP synthesis (Figs 5 and EV3A). Purine breakdown products such as xanthine and cAMP, as well as levels of the precursor glutamine, decreased after mAD. Similarly decreased was another purine derivative, allantoin (Fig 5D), which is produced non-enzymatically from uric acid by reactive oxygen species (Grootveld & Halliwell, 1987). Pyrimidine pathway intermediates such as 2-deoxyuridine were increased after mAD in patients and controls (Fig 5D), potentially replenishing the TCA cycle. Taken together, these results indicate that mAD modifies considerably the metabolism of both PEO patients and healthy controls, boosting the energy carrier pool and utilizing alternative energy sources such as BCAAs. The similar major metabolic changes in PEO patients and controls suggested that the metabolic inflexibility was pronounced only in affected muscle fibers, which are unable to oxidize non-glucose carbon sources even if glucose was low.

### Mice do not show a similar acute response to high-fat diet

The patient trial reported here was based on our previous mouse study (Ahola-Erkkila et al, 2010): Deletor with MM (carrying the same Twinkle mutation as our patients 2–4) showed significantly delayed progression of MM after ketogenic diet (KD) of 10 months. Because of the unexpected acute adverse response of the PEO patients to the diet, we asked whether the mice also had an acute response of muscle degeneration, which had remained undetected, followed by improvement. Here, we applied 1 week of ketogenic diet to Deletor mice. We detected a clear increase in plasma ketone levels in these mice (P-OHButyr ND $0.19 \pm 0.06$ mmol/l and KD $0.78 \pm 0.14$ mmol/l), but found no signs of muscle degeneration, neither by histological examination nor by blood analysis of CK values (ND $436 \pm 386$ U/l and KD $137 \pm 76$ U/l). This result indicated that mice and humans with MM have different responses to high-fat, low-carbohydrate diet, indicating species-specific flexibility in coping with mitochondrial dysfunction and changing diets.

## Discussion

Mitochondria are the cellular hub for nutrient oxidation and contribute also to cellular biosynthetic reactions, linking nutrient intake to cellular energy metabolism and growth (Nikkanen et al, 2016). However, the effects of different diets for mitochondrial disease progression are unexplored. We report here a pilot study enrolling PEO patients and healthy control subjects on a high-fat, low-carbohydrate modified Atkins diet. mAD resulted in a uniform response with subacute progressive damage of muscle fibers, especially RRFs, in all of our MM patients, regardless of their genotype or age. The finding is important, as it (i) indicates directly that a diet can modify mitochondrial disease progression; (ii) suggests that a short mAD could decrease RRF number in a wide range of skeletal muscles; and (iii) shows that Atkins diet—a common weight-loss method—may cause acute muscle damage in a specific population subgroup, raising the question whether an extreme dietary scheme could contribute to manifestation of subclinical MM.

Our patients reached mAD-induced ketosis similarly as the controls and showed increased serum BCAAs and other alternative energy transducing pathways, indicating partially normal metabolic

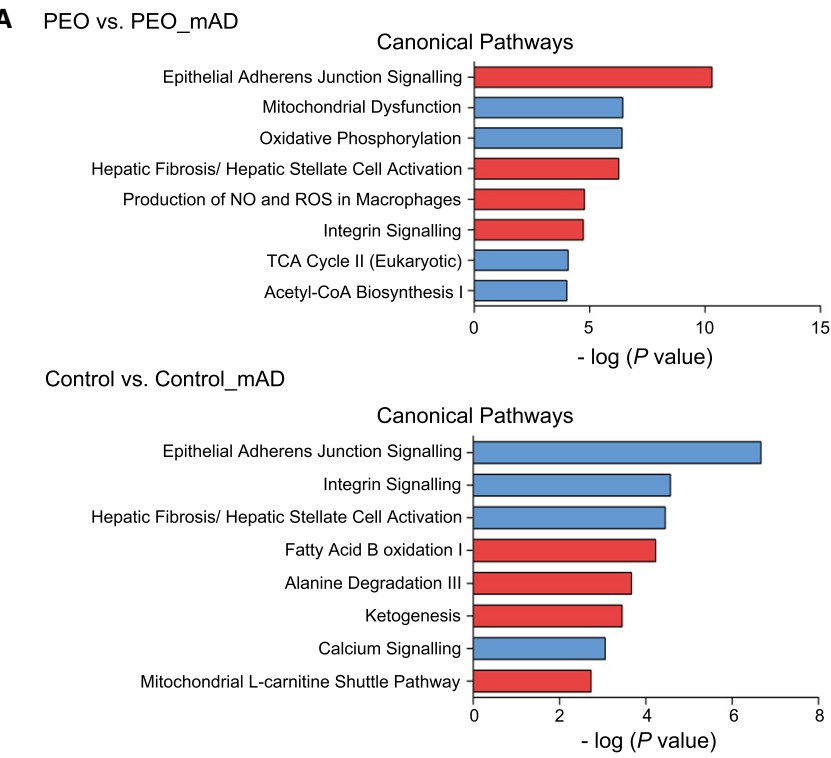

**B**

**PEO**

| Ingenuity Canonical Pathways | *P* value | Molecules |
|---|---|---|
| Epithelial Adherens Junction Signalling | 5.01E-11 | MYH6,ARPC1B,MYL6,TGFBR3,ACTA2,MYH11,TUBB,MYH7B,TGFBR2,SORBS1,MRAS,MYL4,TUBA1C,ACTG2,ACTC1,ACTN1,TUBB3,TUBB4B,RRAS,ACTB,ACTN2,PTPRM,TUBA1B,MYL9,TUBA1A,TUBB6,WAS,MYH9,ZYX,ARPC4 |
| Mitochondrial Dysfunction | 3.63E-07 | SDHA,ATP5J,NDUFAF1,NDUFV1,NDUFA9,UCP2,COX6A1,ATP5A1,COX10,TRAK1,NDUFB8,UQCRB,NDUFS3,PDHA1,NDUFS8,ATP5G1,UQCRC2,CYC1,COX5A,CYCS,VDAC1,UQCRC1,ATP5F1,PINK1 |
| Oxidative Phosphorylation | 3.98E-07 | SDHA,ATP5J,NDUFV1,NDUFA9,COX6A1,COX10,ATP5A1,NDUFB8,UQCRB,NDUFS3,NDUFS8,ATP5G1,UQCRC2,CYC1,COX5A,CYCS,UQCRC1,ATP5F1 |
| Hepatic Fibrosis / Hepatic Stellate Cell Activation | 5.50E-07 | IGFBP4,MYH6,VCAM1,CTGF,MYL6,TNFRSF1A,ACTA2,IFNGR1,MYH11,MYH7B,TGFBR2,MYL9,LY96,CCL2,TIMP1,IGFBP3,CD14,MYH9,MYL4,SERPINE1,TNFRSF1B,MMP9,TIMP2 |
| Production of NO and ROS in Macrophages | 1.66E-05 | PIK3C2B,APOE,JAK1,RHOC,TNFRSF1A,MAP3K1,PPP1R3A,IFNGR1,PCYOX1,SPI1,APOC1,FOS,LYZ,RHOG,RHOB,PPP2R3A,PPM1J,CYBA,CYBB,S100A8,IRF8,TNFRSF1B,SIRPA,APOD |
| Integrin Signaling | 1.86E-05 | RAC2,ARPC1B,ARHGEF7,ACTA2,RHOG,RHOB,MRAS,CAV1,ACTG2,TSPAN4,ACTC1,ITGB5,ACTN1,PIK3C2B,RRAS,RHOC,ACTN2,ACTB,TTN,MYL9,ITGB2,MYL12A,WAS,ZYX,ARPC4 |
| TCA Cycle II (Eukaryotic) | 8.51E-05 | SDHA,SUCLA2,CS,SUCLG1,DLD,MDH1,FH |
| Acetyl-CoA Biosynthesis I (Pyruvate Dehydrogenase Complex) | 9.77E-05 | PDHA1,DLAT,DLD,PDHB |

**Control**

| Ingenuity Canonical Pathways | *P* value | Molecules |
|---|---|---|
| Epithelial Adherens Junction Signalling | 2.14E-07 | MYL9,MYH6,MYL6,ACTB,MYL5,ACTA2,ZYX,ARPC3,ACTG2,MYH11,ACTC1,TUBA1B,ACTN1 |
| Integrin Signaling | 2.69E-05 | MYL9,RHOD,ACTB,ACTA2,MYL5,ARPC3,ZYX,MYLK,ACTG2,BCAR3,ACTC1,ACTN1 |
| Hepatic Fibrosis / Hepatic Stellate Cell Activation | 3.55E-05 | MYL9,COL1A2,MYH6,CTGF,MYL6,TNFRSF1A,ACTA2,MYL5,MYH11,COL3A1 |
| Fatty Acid β-oxidation I | 5.89E-05 | HADHB,IVD,ACAA2,ACSL1,HADHA |
| Alanine Degradation III | 2.14E-04 | GPT,GPT2 |
| Ketogenesis | 3.55E-04 | HADHB,HMGCS2,HADHA |
| Calcium Signalling | 8.71E-04 | MYL9,CALM1,CALR,MYH6,MYL6,ACTA2,MYL5,MYH11,ACTC1 |
| Mitochondrial L-carnitine Shuttle Pathway | 1.86E-03 | CPT1B,CPT2,ACSL1 |

**Figure 4.  Transcriptomic analysis of PEO muscle, before and after mAD.**

A  Most significantly altered transcriptomic pathways in PEO muscle (*n* = 4) and in control subjects muscle after mAD (*n* = 8). Blue color represents down-regulated and red color represents up-regulated pathways.

B  Genes, the expression levels of which formed the pathway results in (A). Statistical test: Benjamini-Hochberg multiple testing.

response to low-carbohydrate diet of PEO patients. An increase in plasma BCAA levels has been reported as a response to high-protein diet in a study with human subjects with isocaloric diet design (Chiu *et al*, 2014). However, the subacute degenerative response of RRFs, "selective rhabdomyolysis", and widespread down-regulation of the whole mitochondrial oxidative pathway expression indicated that RRFs are especially dependent of carbohydrates and are not able to oxidize ketones, amino acids, or lipids for ATP production. Indeed, MM patients accumulate lipids, especially inside RRFs, which is a typical diagnostic finding for the disease. Our MRI data verified a significant accumulation of fat in the patients' muscle and liver. Lipid oxidation is strictly controlled by redox status, especially NADP/NADPH ratio, and reduced availability of $NAD^+$, demonstrated in MM mice (Khan *et al*, 2014), could underlie the metabolic block.

Iatrogenic induction of muscle degeneration has been suggested as a therapeutic strategy for MM patients carrying primary heteroplasmic mtDNA mutations (Clark *et al*, 1997). The rationale of this idea is based on the finding that muscle satellite cells do not carry mtDNA deletions, and their activation by damage would induce their fusion to existing muscle fibers, replenishing them with wild-type mtDNA. To purposefully induce muscle damage, previous reports utilized bupivacaine hydrochloride injections to patient muscle (Clark *et al*, 1997) or strenuous exercise (Taivassalo *et al*, 1999; Murphy *et al*, 2008). Our short dietary intervention of 1–1.5 weeks turned out to be an efficient way to stress the metabolism of the whole-body musculature of MM patients, especially targeting the damage to the most affected RRFs. This raised the question whether such an induction could actually be beneficial by inducing satellite cell fusion. We did not find signs of satellite cell activation in the acute lytic phase, but the improving muscle strength in all patients after 2.5 years, especially in the youngest patients, suggests that satellite cell induction could occur in the recovery phase.

Our patients with either single or multiple mtDNA deletions, that is, sporadic or secondary mtDNA mutations, responded strikingly uniformly to mAD, both qualitatively and chronologically. The symptom progression, ascending from lower extremities to back muscles, arms, and neck muscles, was similar in all, as was the timing of the lytic response, based on increase of serum CK and myoglobin. The remarkable similarity of the responses despite their different genetic backgrounds suggests a uniform molecular pathophysiology leading to manifestation of MM/PEO. Mitochondrial disorders typically manifest with variable phenotypes, even in patients with the same genetic lesion. Our findings emphasize the importance of phenotypic uniformity of mitochondrial disease

patient groups in therapy trials in general—a factor perhaps more important than uniformity of genotypes.

No knowledge of mAD effects in muscle exists, despite it being a popular weight-loss scheme in the normal population and also used in treatment of severe childhood epilepsies and mitochondrial encephalopathies (for a review, see Sharma & Jain, 2014) (Panetta *et al*, 2004; Kang *et al*, 2007; Joshi *et al*, 2009). These case reports have shown a decreased epileptic attack frequency during ketogenic diet, but the diseases have progressed typically with a fatal course, rendering the results of the true effects inconclusive. Typically, mAD effects on muscle enzymes were not reported, even when used in childhood mitochondrial disease. Our results suggest that CK should be followed up when utilizing high-fat, low-carbohydrate diet in treatment of mitochondrial disorders.

Our previous study on Deletor mice with MM demonstrated that long-term high-fat, low-carbohydrate diet strongly induced mitochondrial biogenesis and cured the ultrastructural changes of MM (Ahola-Erkkila *et al*, 2010). The Deletors mimic PEO patients in having COX-negative, SDH-positive muscle fibers, although their RRFs do not carry paracrystalline inclusions (Tyynismaa *et al*, 2005). Therefore, the acute response of our PEO patients in a similar study was unexpected. We here performed a short-term ketogenic diet for Deletor mice, to ask whether the mice had a similar acute response that had remained undetected. However, no signs of muscle degeneration or CK elevation were found in Deletors. These findings indicate that the flexibility of mouse muscle to adapt to different diets is greater than in humans, even when suffering from mitochondrial myopathy. Furthermore, the results emphasize the importance of small pilot trials to follow promising treatment results in mouse models.

mAD has been suggested as a treatment for obesity and T2D (for a review, see Paoli *et al*, 2013). Our study did not indicate negative side effects in healthy subjects. The transcriptome assay indicted mitochondrial biogenesis induction and boosted oxidative phosphorylation. Control subjects showed a rapid metabolic shift to ketosis, induction of oxidative metabolism in the muscle, and secretion of alternative energy sources in the blood. However, a long-term significant increase in plasma BCAA levels could be a risk factor for developing insulin resistance and T2D (Newgard *et al*, 2009; Wang *et al*, 2011; Newgard, 2012). The healthy subjects also showed slightly reduced plasma triglyceride levels, but cholesterol, insulin, or glucose levels remained unchanged. These findings are in agreement with previous reports of low-carbohydrate diets in human subjects (for a review, see Naude *et al*, 2014).

In conclusion, our results indicate that nutrition can drastically modify progression of MM and that RRFs are dependent on

**Figure 5.  Metabolomic analysis of plasma on normal diet and after mAD.**

A    PEO patients, relative amino acid levels on normal diet (PEO), immediately after finishing mAD (PEO_mAD), and 1 month after mAD.

B    Control subjects, relative amino acid levels on normal diet (Control) and immediately after finishing mAD (Control_mAD).

C    Relative values of plasma metabolites in the creatine pathway, compared to controls on ND. Right: illustration of major metabolites and pathway of creatine synthesis.

D    Relative values of plasma purine and pyrimidine degradation pathway intermediates, compared to controls on ND. Right: illustration of major metabolites and synthesis of pyrimidines and purines.

Data information: Values in all charts represent mean of relative values to untreated controls mean (dashed line) ± SEM. Statistical analyses: Student's *t*-test (two-tailed). *$P < 0.05$, **$P < 0.01$, ***$P < 0.001$. Blue, decrease after mAD in PEO; red, increase after mAD in PEO. Abbreviations: GAA, guanidinoacetic acid; SAM, S-adenosylmethionine; SAH, S-adenosylhomocysteine; MTHF, methyltetrahydrofolate; DHF, dihydrofolate; dUMP, deoxyuridylate; dTMP, deoxythymidylate; dU, 2-deoxyuridine; AIBA, aminoisobutyric acid; cAMP, cyclic adenosine monophosphate; GMP, guanosine monophosphate; IMP, inosine monophosphate; CK, creatine kinase.

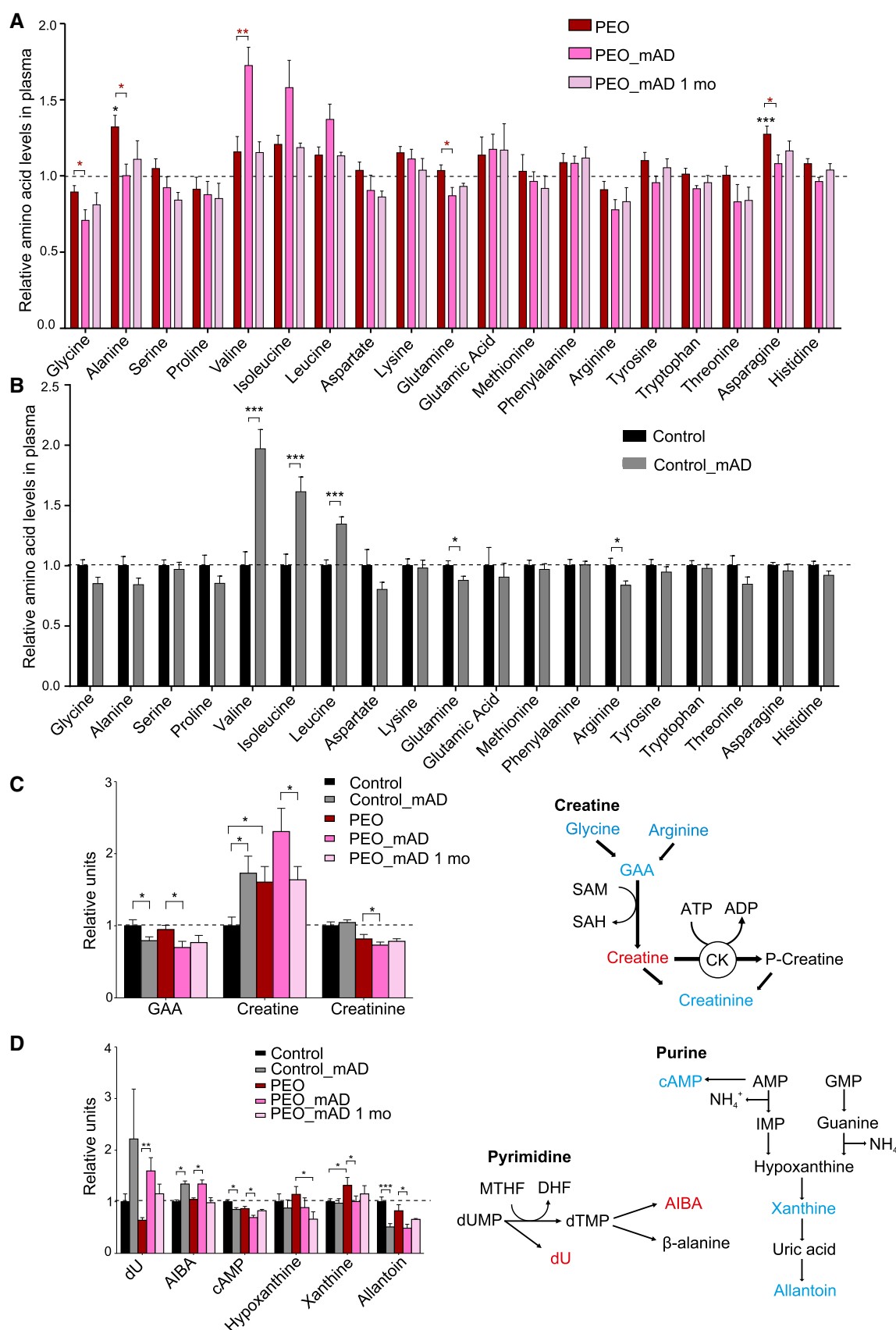

**Figure 5.**

carbohydrate supply. Furthermore, our results show that mAD may induce muscle cell damage in a small population group with subclinical mitochondrial disease. Based on our results, muscle symptoms and CK values should be closely monitored if low-carbohydrate diet is used as treatment for mitochondrial disease patient. The uniform response of our PEO patients to mAD indicates existence of a yet-unknown metabolic block in adult-onset MM. The exciting potential of dietary intervention as a treatment of MM/PEO, aiming to target and reduce the number of RRFs, requires further investigation.

# Materials and Methods

All of the subjects gave their written informed consent to participate in the study. The study was undertaken according to the Helsinki declaration. Helsinki University Hospital ethics review board approved the study. National Animal Experiment Board, Finland, approved animal work, and all experiments were done in accordance with good practice of handling laboratory animals.

## Diet

The subjects filled food diaries for 7 days prior to the diet study, which was used as a basis to calculate their normal daily caloric intake and to design the individual isocaloric diets of the trial. Table EV1 and Fig EV1A describe the detailed diet compositions. The subjects were first enrolled on standardized normal diet (ND) for 2 weeks, after which they descended gradually to mAD: One meal per day was changed into mAD. All subjects were to continue on mAD for a total of 4 weeks.

## Muscle strength and performance analyses

Muscle strength of the patients was analyzed before and after mAD by Medical Research Council score (Hahn *et al*, 1996), grip strength by Martin Vigorimeter, and isometric muscle strength with "Good strength" analyzer (Metitur Oy, Finland) (Era *et al*, 1992), including elbow flexion and extension, lower leg extension, and grip strength on both sides. In addition, the patients were tested for balance (standing with one foot), walking for 6 min, static strength of back muscles, and repetitive strength of stomach and upper limb muscles. In addition, a "muscle symptom score" was determined by questionnaire, designed to assess the main symptoms of PEO patients, and RAND-36 questionnaire was utilized to assess their general quality of life (Hays *et al*, 1993). In "muscle symptom score" questionnaire, patients were asked to score their muscle pain in scores of "not at all, mild, moderate, greatly, very greatly" in "rest, activity, numbness, muscle tiredness, muscle weakness, muscle cramps". Patients were also asked to use the same scoring to identify the muscles that have symptoms "calf and leg, thigh, upper limbs, back and body, neck, head, eyelids and muscles moving of the eyes".

## Laboratory examinations

Laboratory examinations were taken after overnight fasting: blood count, alanine aminotransferase, asparate aminotransferase, glutamyl transferase, cholesterol, high-density lipoprotein, low-density lipoprotein, triglycerides, creatine kinase (CK), lactate, pyruvate, creatinine, urea, FGF-21, glucose, insulin, ketones, and blood gases. FGF-21 was analyzed using BioVendor human FGF21 ELISA kit.

## Spiroergometry

Spiroergometry with blood gas, lactate, and ammonium analysis was performed as in Mustelin *et al* (2008). The initial workload was 30–40 W for patients, 40 W for control women, and 50 W for control men. $HbO_2$, $pCO_2$, lactate, and $NH_4^+$ concentrations were determined during rest, light exercise, and maximum exercise, as well as 2, 4, 6, 10, 20, and 30 min after exercise.

## Magnetic resonance imaging (MRI) and spectroscopy (MRS)

MRI/MRS experiments were performed on a 3.0-Tesla clinical imager (Verio; Siemens). Point-resolved spectroscopy (PRESS) sequence was used for hepatic MRS. Liver spectra were analyzed by jMRUI v3.0 software, with the AMARES algorithm (Vanhamme *et al*, 1997), and fat content was expressed as 100× methylene intensity divided by the sum of methylene and water intensities. Abdominal MRI was analyzed from 8 cm above to 8 cm below the L4/5 lumbar intervertebral disks. Visceral (VAT) and subcutaneous (SAT) fat areas were determined using a region growing method in SliceOmatic v4.3 image analysis program (Tomovision). Muscle MRI was analyzed from the middle of the soleus muscle. The intensities of intramyocellular lipids (IMCL), extramyocellular lipids (EMCL), and creatine (Cr) signals were determined from water-suppressed spectra using LCModel v6.3 software. IMCL and EMCL contents were expressed as ratio to Cr intensity.

## Muscle sampling and histological analysis

A Bergström needle biopsy from *vastus lateralis* muscle was taken before and after mAD (Fig 1A). The muscle samples for DNA, RNA, and protein analyses were snap-frozen in liquid nitrogen, and samples for histology were snap-frozen in liquid nitrogen-cooled isopentane. All samples were stored at −80°C. Muscle samples for electron microscopy (EM) were fixed in glutaraldehyde. Eight-micrometer-thick frozen muscle sections were tested for cytochrome c oxidase (COX)/succinate dehydrogenase (SDH) activities by standard histochemical activity assay. A total of 250–300 cells per patient were counted to assess COX-negative/SDH-positive cells. Immunohistochemistry was performed on 8-μm frozen sections for p62 (dilution 1:500, Santa Cruz Biotechnology, #sc-28359/D-3), cleaved caspase-3 (1:150 dilution, Cell Signaling, #9664), LC3 (1:50 dilution, nanoTools Antikörpertechnik GmbH & Co. KG, #0231-100/LC3-5F10), and mitochondrial marker antigen (dilution 1:10, Biogenex, #MU213-UC/clone 113-1). Peroxidase stainings were done with VECTASTAIN Elite ABC HRP Kit (Vector Laboratories) for caspase and LC3 and with Multimer kit Ultraview Universal DAB detection system (Ventana Medical Systems). Transmission electron microscopy sample preparation and analysis was done as previously described (Ahola-Erkkila *et al*, 2010). A total of 70–190 cells per patient were counted from ultrathin sections, stained with toluidine

blue, to determine the proportion of muscle fibers showing signs of cell death.

## RNA analysis

Total muscle RNA was extracted with standard TRIzol and chloroform method and purified with RNA purification kit (RNeasy; Qiagen). A total of 250 ng of total RNA from four PEO patients and eight controls was used for global gene expression analysis on the HumanHT-12 v4 Expression BeadChip (Illumina) with TotalPrep RNA Labeling Kit (Ambion). Chipster v2.1.0 software was used for quantile normalization (http://chipster.github.io/chipster/). Data were further filtered by coefficient of variation (standard deviation/mean; 50% of all present genes were filtered out). To determine the differently expressed genes between PEO or controls on ND and after mAD, paired empirical Bayes statistical test was applied together with the Benjamini–Hochberg *P*-value adjustment method. For further analysis, the transcripts with adjusted *P*-value < 0.05 for differential expression between PEO on standard diet and PEO on Atkins diet were selected regardless of fold change and subjected to pathway analysis using Ingenuity Pathway Analysis (IPA; Ingenuity® Systems; www.ingenuity.com). All genes with log2 fold change > 0.26 and < −0.26 regardless of adjusted *P*-value between controls on standard diet and controls on Atkins diet were used for IPA.

## DNA analysis

DNA was extracted from the muscle samples using standard phenol–chloroform extraction and ethanol precipitation and mtDNA copy number was analyzed by qPCR, with primers from the mtDNA gene for 12S rRNA and nuclear gene for amyloid-beta precursor protein (APP), as previously described (Tyynismaa *et al*, 2012). mtDNA deletion analysis was done using a quantitative PCR method and the primers described in He *et al* (2002) with iQ™ Custom SYBR® Green Supermix (Bio-Rad).

## Protein analysis

Protein extracts were separated in 12% SDS–gel electrophoresis, using antibodies against p62 (dilution 1 μg/ml, Abcam®, ab56416) and pax7 (dilution 1 μg/ml, Abcam®, ab34360). Beta-tubulin antibody (dilution 1:1,000, Cell Signaling Technology®, #2128) was used as a loading control, and membranes were imaged using Bio-Rad ChemiDoc™ XRS+ with Image Lab software.

## Metabolomic analysis

Plasma was separated immediately after sampling and stored at −140°C. Metabolites were analyzed with ACQUITY UPLC-MS/MS system (Waters Corporation) as in Khan *et al* (2014). The column was 2.1 × 100 mm Acquity 1.7um BEH amide HILIC column and the detection system Xevo® TQ-S tandem triple-quadrupole mass spectrometer (Waters Corporation). In order to analyze differences among groups, univariate analysis *T*-test2 was performed on autoscaled data. In order to explain the separation among groups, unsupervised multivariate analysis, principal component analysis (PCA), was performed. Non-transformed data were mean-centered and autoscaled. Five components were used in all the PCA models.

**The paper explained**

**Problem**
Mitochondrial myopathy is a progressive genetic disorder with no treatment. Animal studies suggested that ketogenic high-fat, low-carbohydrate diet ameliorates the disease.

**Results**
Five adult patients with mitochondrial myopathy followed a high-fat, low-carbohydrate "modified Atkins" diet, which resulted in progressive muscle pain and signs of muscle damage in all patients within 2 weeks. This diet especially caused lysis of the ragged-red fibers (RRFs), the hallmarks of mitochondrial myopathy. Such changes were independent of patient genotype. The changes were accompanied by subtle improvement of muscle strength in a 2.5-year follow-up. Healthy subjects did not show such changes.

**Clinical impact**
mAD can induce muscle damage in mitochondrial disease patients. The specific targeting of RRFs indicates a potential to decrease RRF number by short dietary intervention. Evidence suggests that an Atkins diet could provoke muscle damage in a population subgroup, that is, people with subclinical mitochondrial disease.

Dendrogram was plotted on normalized data using Ward's linkage clustering algorithm and Pearson's correlation similarity measure and visualized as heatmap.

## Short-term ketogenic diet for Deletor mice

Twenty-month-old male Deletor mice (Tyynismaa *et al*, 2005), littermates in congenic C57Bl6 background, were randomized and fed ketogenic or normal diet as in Ahola-Erkkila *et al* (2010) [ketogenic diet *n* = 5, normal diet *n* = 3; diets: high-fat diet D05052004 (89.5 kcal% fat, 0.1 carbohydrate, 10.4 protein); control diet D05052002 (11.5 kcal% fat, 78.1 carbohydrate, 10.4 protein; Research Diets Inc.)] for 7 days, after which the animals were fasted for four hours and then killed and blood was collected by cardiac puncture. Serum samples were analyzed for CK and ketone bodies, to assess muscle damage and success of the diet. *Quadriceps femoris* muscle samples were fixed in glutaraldehyde and and ultrathin sections were stained with toluidine blue and analyzed by light microscope to assess cell death as in Ahola-Erkkila *et al* (2010). Samples were numbered to ensure blinding of the investigator.

**Expanded View** for this article is available online.

## Acknowledgements

The authors wish to thank Markus Innilä and Anu Harju for technical assistance, Mugen Terzioglu and Mervi Kuronen for experimental expertise, and Christopher Carroll for editing figures and proofreading the manuscript. The authors also wish to thank the following institutions for funding: Sigrid Juselius Foundation, Jane and Aatos Erkko Foundation, European Research Council, Academy of Finland, University of Helsinki and Helsinki University Central Hospital (for A.S.); and Biomedicum Helsinki Foundation, Helsinki Biomedical Graduate School, University of Helsinki, Waldemar von Frenckells stiftelse, Finnish Cultural Foundation, and Alexander von Humboldt Foundation (for S.A.).

## Author contributions

SA, MA, PI, and AS designed the study, and SA, MA, and AS wrote and edited the manuscript. SA analyzed the data and performed most laboratory analyses, mouse study, and microscopy. MA and PI recruited and examined patients and participated in the analysis of the data. SN and KHP designed the diets, instructed patients during the study, and performed calorimetry analyses. NU performed muscle strength and exercise tests. JB analyzed metabolomic data and helped with manuscript preparation. VV performed metabolomic analyses. NL and AH performed MRI analyses. TM and PP did spiroergometry analyses and helped with manuscript preparation. AS supervised the study, recruited patients, and analyzed and interpreted the data.

## Conflict of interest

The authors declare that they have no conflict of interest.

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
