## [Review Process File · EMBO Molecular Medicine]

Modified Atkins diet induces subacute selective ragged-red-fiber-lysis in mitochondrial myopathy patients

Sofia Ahola, Mari Auranen, Pirjo Isohanni, Satu Niemisalo, Niina Urho, Jana Buzkova, Vidya Velagapudi, Nina Lundbom, Antti Hakkarainen, Tiina Muurinen, Päivi Piirilä Kirsi Pietiläinen and Anu Suomalainen

Corresponding author: Anu Suomalainen, University of Helsinki

Review timeline:

Submission date:	10 May 2016
Editorial Decision:	09 June 2016
Revision received:	28 July 2016
Editorial Decision:	08 August 2016
Revision received:	10 August 2016
Accepted:	12 August 2016

Transaction Report:

Editor: Roberto Buccione

1st Editorial Decision

09 June 2016

Thank you for the submission of your manuscript to EMBO Molecular Medicine. We are sorry that it has taken longer than we would have liked to get back to you on your manuscript.

As you will see the three reviewers are very positive, although reviewer 3 is slightly more reserved.

In brief, while publication of the paper cannot be considered at this stage, we would be pleased to consider a revised submission, with the understanding that the Reviewers' concerns must be addressed including further experimentation at the very least to address the concerns raised by reviewer #3 related to autophagy assessment and FGF21, and if possible the lack of muscle biopsy data in the long term follow up. Eventual acceptance of the manuscript will entail a second round of review.

We remind you that it is EMBO Molecular Medicine policy to allow a single round of revision only and that, therefore, acceptance or rejection of the manuscript will depend on the completeness of your responses included in the next, final version of the manuscript.

As you know, EMBO Molecular Medicine has a "scoping protection" policy, whereby similar findings that are published by others during review or revision are not a criterion for rejection. However, I do ask you to get in touch with us after three months if you have not completed your revision, to update us on the status. Please also contact us as soon as possible if similar work is published elsewhere.

Please note that EMBO Molecular Medicine now requires a complete author checklist (<http://embomolmed.embopress.org/authorguide#editorial3>) to be submitted with all revised manuscripts. Provision of the author checklist is mandatory at revision stage; The checklist is designed to enhance and standardize reporting of key information in research papers and to support reanalysis and repetition of experiments by the community. The list covers key information for figure panels and captions and focuses on statistics, the reporting of reagents, animal models and human subject-derived data, as well as guidance to optimise data accessibility.

We now mandate that all corresponding authors list an ORCID digital identifier. You may do so through our web platform upon submission and the procedure takes < 90 seconds to complete. We also encourage co-authors to supply an ORCID identifier, which will be linked to their name for unambiguous name identification.

Please carefully adhere to our guidelines for authors (<http://embomolmed.embopress.org/authorguide>) to accelerate manuscript processing in case of acceptance.

I look forward to seeing a revised form of your manuscript as soon as possible.

***** Reviewer's comments *****

Referee #1 (Comments on Novelty/Model System):

Small clinical trial of adults with mitochondrial myopathy on the ketogenic diet with associated clinical course, biochemistry and histopathological correlates. Implications for future therapy for these disorders. Human subjects were enrolled according to standard ethical procedures as stated.

Referee #1 (Remarks):

This manuscript depicts a small clinical trial utilizing the modified Atkins diet (MAD) for adults with mitochondrial myopathy (MM). The article points out that the modified Atkins diet has been used in mitochondrial epilepsy patients with success, but has not been studied in MM. The findings are quite interesting, in that the adults had muscle cramping and myolysis of ragged red fibers within a short time on the diet. The oldest patient stayed on the diet only 4 days; shorter compared to the others and perhaps due to more symptoms given that RRF increase with age (?). The subjects were assessed 2 years later and found to have some improvements, leading to the conclusion that this was the effect of the MAD for about one week 2 years prior; this may be a stretch without other information about the interval health and interventions. The authors draw a conclusion that the MAD is similar to bupivacaine or exercise in terms of bringing in satellite cells to recruit for improved muscle strength and improved heteroplasmy in the case of mtDNA mutations. The biochemical testing revealed elevated CPK and the authors recommend that mitochondrial patients on the KD or MAD have CPK tested during the diet for monitoring purposes based on this trial. This manuscript is very important in terms of overall approach to treatment of these disorders; currently the field is interested in probing into diet and supplements for treatment.

Referee #2 (Remarks):

The manuscript by Ahola et al presents striking and very interesting results. In short, the authors show that a ketogenic "Atkins" diet has similar overall effects on controls and patients with mitochondrial myopathy (investigated in many different ways). But they also show that the diet also specifically damages ragged red fibers. In fact, during the trial, patients started to sustain muscle pain and other signs of damage and the trial had to be stopped. Interestingly, the patients recovered well and might (really just might) have gained some muscle strength in the process.

These observations lead to several interesting conclusions:

- 1) Affected muscle fibres in MM patients are characterized by an acute inability to gain ATP in any other way than glycolysis.
- 2) Diets that can in fact alter metabolism should not be used indiscriminately, even with apparently healthy individuals (because of possible underlying conditions).
- 3) And (possibly) damaging ragged red fibres could stimulate muscle regeneration from satellite cells. However, the current data is far from demonstrating this convincingly. Thus, the possibility of the success of such an approach remains highly conjectural. Which is not a criticism of the interest of the data presented.

Minor Criticisms:

- 1) There should be a more extensive discussion that compares and contrasts the findings in mice and in people.
- 2) There are numerous small syntactical, grammatical and lexicographic errors. These are too numerous to cite all here. I provide three examples (out of many that should be checked and corrected).
 - a. Abstract, line 2: "adult age" is not a proper turn of phrase in English.
 - b. Abstract, penultimate line: degenerate is an intransitive verb; you cannot degenerate something.
 - c. End of page 4: you write decent when you mean descent; and, in any case, even "descent to mAD" is not immediately understandable. One speaks of descent into madness or into darkness, not so much into a diet (but maybe you meant that such a diet is hell).

Referee #3 (Comments on Novelty/Model System):

This work takes advantage of advanced techniques, including metabolomic and transcriptomic analysis.

The study is one of the first example of in vivo study on mitochondrial myopathy patients and suggest a new possibility to ameliorate the disease by short administration of high-fat, low carbohydrate diet.

Referee #3 (Remarks):

In this paper, Ahola et al describe the effects of a high fat low carbohydrates modified Atkins diet on mitochondrial myopathy patients. This study stems from a previous study from the same group showing that a ketogenic diet delays the progression of the disease a mouse model of mitochondrial myopathy, possibly by inducing mitochondrial biogenesis. The authors characterize the effects of mAD in the patients, and found that mAD induces acute muscle damage in PEO patients after only 7-10 days by triggering both apoptosis and necrosis in affected fibres, but not autophagy. A mild increase in muscle strength is observed up to 2.5 years after the treatment. In addition, the authors carry out an extensive characterization of the metabolic and transcriptomic changes accompanying the mAD regime in patients and controls. The main differences they observe are related to the inability of RRF in patients' muscles to use fatty acids and ketones stimulated by mAD as an energy source and thus rely more heavily on glucose, as suggested by increased lactate levels, and accumulation of fat in muscles and livers. Surprisingly, no activation of mitochondrial biogenesis is observed in PEO patients after mAD, including upregulation of beta-oxidation-related genes. Finally, no major differences were observed in the metabolomic survey between controls and patients, although the normalization of some metabolites, such as alanine, which is a biomarker of mitochondrial disease, in PEO patients is observed.

The paper is interesting as it extensively describes metabolic and transcriptomic changes in PEO patients in both ND and mAD regime, and may open new possibilities for the treatment of mitochondrial disease, but it remains very descriptive and most of the findings should be supported by more convincing evidence, as detailed below.

A first criticism is that a muscle biopsy in the long-term follow up is missing. The mild improvement can obviously be related to mAD-induced changes and possibly amelioration of the OxPhos deficiency, but this is not proven by the data provided.

A second concern is about the design of the study. The authors stopped the mAD administration in patients after one week, but continued it in the controls over 4 weeks, which make the results difficult to interpret to some extent. For instance the authors describe an increase in mitochondrial biogenesis in controls but not in patients. This is opposite to what is observed in mice, but it can be due to the shorter duration of mAD in the patients.

An additional point is that the characterization of mAD induced damage should be improved. In particular, the staining with the anti p62 antibody is Supplementary figure 1K is not very convincing as the pattern seems rather diffuse and does not show the expected pattern (puncta). In addition, it's risky to draw conclusions on autophagy based on a single marker, although p62 levels are convincingly increased in the western blot. However, p62 can be increased transcriptionally and it's not necessarily related to autophagy. Additional markers of autophagy (such as LC3) should be added to support this finding.

Other points:

There is no analysis of FGF21, which is probably the most reliable biomarker of the disease, is reported. Have the authors measured it in ND and mAD? Have they followed-up FGF21 in their long-term analysis of the patients?

Figure 1 E and H: please add dimension bars

Supplementary Figure 1I: Is this a relative quantification of mtDNA? I noticed that at least in some of the patients there is a trend towards a reduction in mtDNA copy number. In addition, one of them has a much higher starting value and a clear reduction upon mAD. I wonder if this is the same patient showing very high levels of PAX7 and p62 in figure 1N.

In figure 1L,M the quality of the pictures is not particularly good, especially for the staining with the anti-SDH antibody. Why is there a strong stain of the fibres borders with Caspase 3? Can you improve them?

Figure 2 is rather confusing, especially for lactate. Although an increase in lactate levels upon mAD can be appreciated, it is quite difficult to compare the other groups (e.g. controls and PEO in normal diet).

Finally, do the authors think that repeated periods of mAD can be more effective in mitochondrial myopathies?

1st Revision - authors' response

28 July 2016

Referee #1 (Comments on Novelty/Model System):

Small clinical trial of adults with mitochondrial myopathy on the ketogenic diet with associated clinical course, biochemistry and histopathological correlates. Implications for future therapy for these disorders. Human subjects were enrolled according to standard ethical procedures as stated.

Referee #1 (Remarks):

This manuscript depicts a small clinical trial utilizing the modified Atkins diet (MAD) for adults with mitochondrial myopathy (MM). The article points out that the modified Atkins diet has been used in mitochondrial epilepsy patients with success, but has not been studied in MM. The findings are quite interesting, in that the adults had muscle cramping and myolysis of ragged red fibers within a short time on the diet. The oldest patient stayed on the diet only 4 days; shorter compared to the others and perhaps due to more symptoms given that RRF increase with age (?). The subjects were assessed 2 years later and found to have some improvements, leading to the conclusion that this was the effect of the MAD for about one week 2 years prior; this may be a stretch without other

information about the interval health and interventions. The authors draw a conclusion that the MAD is similar to bupivacaine or exercise in terms of bringing in satellite cells to recruit for improved muscle strength and improved heteroplasmy in the case of mtDNA mutations. The biochemical testing revealed elevated CPK and the authors recommend that mitochondrial patients on the KD or MAD have CPK tested during the diet for monitoring purposes based on this trial. This manuscript is very important in terms of overall approach to treatment of these disorders; currently the field is interested in probing into diet and supplements for treatment.

Authors: We agree with the reviewer's points and truly appreciate the positive and enthusiastic comments of the reviewer, not asking for any changes.

Referee #2 (Remarks):

The manuscript by Ahola et al presents striking and very interesting results. In short, the authors show that a ketogenic "Atkins" diet has similar overall effects on controls and patients with mitochondrial myopathy (investigated in many different ways). But they also show that the diet also specifically damages ragged red fibers. In fact, during the trial, patients started to sustain muscle pain and other signs of damage and the trial had to be stopped. Interestingly, the patients recovered well and might (really just might) have gained some muscle strength in the process.

These observations lead to several interesting conclusions:

- 1) Affected muscle fibres in MM patients are characterized by an acute inability to gain ATP in any other way than glycolysis.
- 2) Diets that can in fact alter metabolism should not be used indiscriminately, even with apparently healthy individuals (because of possible underlying conditions).
- 3) And (possibly) damaging ragged red fibres could stimulate muscle regeneration from satellite cells. However, the current data is far from demonstrating this convincingly. Thus, the possibility of the success of such an approach remains highly conjectural. Which is not a criticism of the interest of the data presented.

Minor Criticisms:

- 1) There should be a more extensive discussion that compares and contrasts the findings in mice and in people.

Authors: Our original data of Deletor mice on long-term ketogenic diet showed considerable improvement of muscle mitochondrial ultrastructure and promoted mitochondrial biogenesis. However, we had not tested the acute effects of mAD in mice, as no apparent symptoms were observed after diet onset. **We however now exposed Deletor mice with mitochondrial myopathy to a short-term mAD of seven days, and studied the effects on plasma CK and ketone bodies (P-OHButyr) as well as muscle cell death by microscopy.** Mice did not have similar RRF-lytic changes in their muscle as the patients, indicating that despite COX-negative, SDH-positive fibers in both species, mice were more flexible in metabolizing nutrients and maintaining muscle structure in an extremely low-carbohydrate – high-fat diet. These data are now added in the manuscript, p15 and p.20 in methods, p.11 results, and p.14 discussion.

- 2) There are numerous small syntactical, grammatical and lexicographic errors. These are too numerous to cite all here. I provide three examples (out of many that should be checked and corrected).

- a. Abstract, line 2: "adult age" is not a proper turn of phrase in English.
- b. Abstract, penultimate line: degenerate is an intransitive verb; you cannot degenerate something.
- c. End of page 4: you write decent when you mean descent; and, in any case, even "descent to mAD" is not immediately understandable. One speaks of descent into madness or into darkness, not so much into a diet (but maybe you meant that such a diet is hell).

Authors: Thank you for careful reading – we have now corrected these errors and also the language of the manuscript **has now been checked by a native English speaker.**

Referee #3 (Comments on Novelty/Model System):

This work takes advantage of advanced techniques, including metabolomic and transcriptomic analysis. The study is one of the first example of in vivo study on mitochondrial myopathy patients and suggest a new possibility to ameliorate the disease by short administration of high-fat, low carbohydrate diet.

Referee #3 (Remarks):

In this paper, Ahola et al describe the effects of a high fat low carbohydrates modified Atkins diet on mitochondrial myopathy patients. This study stems from a previous study from the same group showing that a ketogenic diet delays the progression of the disease a mouse model of mitochondrial myopathy, possibly by inducing mitochondrial biogenesis. The authors characterize the effects of mAD in the patients, and found that mAD induces acute muscle damage in PEO patients after only 7-10 days by triggering both apoptosis and necrosis in affected fibres, but not autophagy. A mild increase in muscle strength is observed up to 2.5 years after the treatment. In addition, the authors carry out an extensive characterization of the metabolic and transcriptomic changes accompanying the mAD regime in patients and controls. The main differences they observe are related to the inability of RRF in patients' muscles to use fatty acids and ketones stimulated by mAD as an energy source and thus rely more heavily on glucose, as suggested by increased lactate levels, and accumulation of fat in muscles and livers.

Surprisingly, no activation of mitochondrial biogenesis is observed in PEO patients after mAD, including upregulation of beta-oxidation-related genes. Finally, no major differences were observed in the metabolomic survey between controls and patients, although the normalization of some metabolites, such as alanine, which is a biomarker of mitochondrial disease, in PEO patients is observed.

The paper is interesting as it extensively describes metabolic and transcriptomic changes in PEO patients in both ND and mAD regime, and may open new possibilities for the treatment of mitochondrial disease but it remains very descriptive and most of the findings should be supported by more convincing evidence, as detailed below.

Authors: We thank the Reviewer of the positive comments. We would like to point out that the metabolomics and transcriptomics samples were taken before onset of the trial and then immediately at cessation of the diet, when the muscles were in the acute myolytic phase. It remains possible that the lack of biogenesis was explained by the acute stress

situation, and biogenesis could have occurred afterwards. However, because the patients consented only to two muscle biopsy samplings, we could not follow this up in human patients.

A first criticism is that a muscle biopsy in the long-term follow up is missing. The mild improvement can obviously be related to mAD-induced changes and possibly amelioration of the OxPhos deficiency, but this is not proven by the data provided.

Authors: we completely agree with the reviewer in all of these comments. However, we were not able to take the third biopsy: the patients consented to two (which are invasive surgical procedures) but not the third, and for the follow-up we had to rely on non-invasive measures. We would like to point out that 1) clinical effects of the diet were remarkable in the acute stage, and 2) the clinical follow-up clearly shows that the acute myolysis did not cause adverse long-term effects

A second concern is about the design of the study. The authors stopped the mAD administration in patients after one week, but continued it in the controls over 4 weeks, which make the results difficult to interpret to some extent. For instance the authors describe an increase in mitochondrial biogenesis in controls but not in patients. This is opposite to what is observed in mice, but it can be due to the shorter duration of mAD in the patients.

Authors: This is a clear misunderstanding. Probably it was not stated clearly enough, but we sampled the matched controls, including their muscle, to match their corresponding patients. As stated in p.5 original ms: "All the control subjects tolerated mAD well, and had no muscle symptoms or elevated plasma CK values, and they all continued on the diet for the full four weeks. However, their sampling scheme, including the second muscle biopsy, was timed similar to their corresponding patients." Therefore the controls are completely appropriate.

We did wonder, as the reviewer, whether we missed an early mouse response in our previous study, and **therefore now performed a short-term mAD for mice.** No CK release or histological signs of muscle damage was found, indicating the mouse muscle is more flexible in its use of different fuel types. **These data are now added in the manuscript, p20 methods, p11 results p14 discussion.**

An additional point is that the characterization of mAD induced damage should be improved. In particular, the staining with the anti p62 antibody in Supplementary figure 1K is not very convincing as the pattern seems rather diffuse and does not show the expected pattern (puncta).

Authors: We agree with the reviewer **and have now improved the stainings.** The biopsy samples were snap frozen for enzyme histology (COX, SDH) and the immunodetection with different antibodies were done also on frozen sections, which always somewhat compromise the muscle morphology and puncta may not be as apparent as for fixed sections. **We however switched our method from immunofluorescence to immunoperoxidase, and have performed new p62, LC3 and cleaved caspase-3 stainings. We utilized also a mitochondrial mass marker, to indicate mitochondrial proliferation.** We do think that the quality of the figures improved by peroxidase method. The new data completely support the previous conclusions. For p62, we show that positive fiber number is increased in PEO muscles and agrees with the Western results, but this was not considerably affected by the diet. **We have now added a new Figure 2 combining the Western and new immunohistology findings.**

In addition, it's risky to draw conclusions on autophagy based on a single marker, although p62 levels are convincingly increased in the western blot. However, p62 can be increased transcriptionally and it is not necessarily related to autophagy. Additional markers of autophagy (such as LC3) should be added to support this finding.

Authors: We agree with the reviewer **and performed now LC3 immunohistology, which is added in Figure 2 H and I.** We show that p62 positive fibers were often also LC3 positive, and that LC3 showed increased mosaic staining in the PEO patients before and after diet, but only after diet we found clear puncta of LC3-positive material in occasional fibers. These findings strongly support the finding of increased presence of p62 and LC3 in PEO, suggesting decreased autophagic flux. This was not affected by the short-term mAD.

Other points:

There is no analysis of FGF21, which is probably the most reliable biomarker of the disease, is reported. Have the authors measured it in ND and mAD? Have they followed-up FGF21 in their long-term analysis of the patients?

Authors: We do agree with the reviewer of usefulness of FGF21 as a biomarker, and have now included the data into the manuscript, **p 5, Figure EV1B.** FGF21 did not significantly change in long-term follow-up.

Figure 1 E and H: please add dimension bars

Authors: OK.

Supplementary Figure 1I: Is this a relative quantification of mtDNA? I noticed that at least in some of the patients there is a trend towards a reduction in mtDNA copy number. In addition, one of them has a much higher starting value and a clear reduction upon mAD. I wonder if this is the same patient showing very high levels of PAX7 and p62 in figure 1N.

Authors: Yes, this is relative mtDNA quantification, against nuclear DNA single-copy gene as described in the methods (page 19). We also clarified this now in Figure legend (Expanded View Figure 1J). P1 has high mtDNA copy number; she is a patient with high % of mtDNA with single mtDNA deletion. However, P2 is the one with highest levels in pax7 and p62 analysis. We did not add these data into the text, because of these being single observations.

In figure 1L,M the quality of the pictures is not particularly good, especially for the staining with the anti-SDH antibody. Why is there a strong stain of the fibres borders with Caspase 3? Can you improve them?

Authors: These stainings were also done on frozen sections, compromising some morphology, as explained above. **We have now performed new cleaved caspase-3 stainings by immunoperoxidase method,** and show that caspase 3 positive fibers exist both pre and post mAD, and that necrotic fibers show no caspase 3 activation, supporting the conclusion that muscle fiber degeneration occurred mostly by necrosis. **These are included as Figure 2A.**

Figure 2 is rather confusing, especially for lactate. Although an increase in lactate levels upon mAD

can be appreciated, it is quite difficult to compare the other groups (e.g. controls and PEO in normal diet).

Authors: We understand the point of the reviewer and to our opinion the confusing appearance was because of overlapping error bars. **We have now used the patient-control-diet code colors to the error bars as well**, and to our opinion, this made the figures more readable. The same revision was done to all subfigures of Figure 3 (previous Fig 2).

Finally, do the authors think that repeated periods of mAD can be more effective in mitochondrial myopathies?

Authors: The reviewer's suggestion is quite interesting, and could be considered in combination with e.g. biogenesis induction strategies. However, this needs to be studied carefully, and unfortunately can not be done with animal models, because of the mouse-human differences in mAD responses. Based on our small trial we stay conservative in our speculation. We had in the original manuscript, in the discussion a sentence: "The exciting potential of dietary intervention as a treatment of MM/PEO, aiming to target RRFs, to stimulate regeneration and improving muscle strength requires further investigation.", which we now modified to be more clear "...aiming to target and reduce the number of RRFs..."

2nd Editorial Decision

08 August 2016

Thank you for the submission of your revised manuscript to EMBO Molecular Medicine.

We have now received the enclosed report from reviewer 3, who also checked the manuscript on behalf of reviewer 2, who was not available. As you will see s/he is now globally supportive and I am pleased to inform you that we will be able to accept your manuscript pending the following final amendments:

- 1) Please comply with the remaining request from the reviewer to amend the manuscript text with respect to figures and the snap freezing protocol. It is also noted that the metabolomics section should perhaps be removed.
- 2) Please clearly demarcate the separation between the two blot strips in Fig. 2c (e.g. with a black line).
- 3) Please include the "The Paper Explained" section in the manuscript

Please submit your revised manuscript within two weeks. I look forward to seeing a revised form of your manuscript as soon as possible.

***** Reviewer's comments *****

Referee #3 (Comments on Novelty/Model System):

The authors provide an unprecedented detailed description of the biochemical, morphological, transcriptional and metabolic changes both in the disease vs. the healthy controls and in mAD vs ND. In addition, they added a short term experiment on mice, highlighting the differences between humans and mice. All this is an important piece of information that will certainly have a profound impact in the field.

Referee #3 (Remarks):

In this new version, the authors improved their manuscript and addressed most of my concerns. It is really a pity that no biopsy is available at 2.5 years, but I understand this raised ethical concerns as not in the original informed consent. However, the authors provide detailed description of the biochemical, morphological, transcriptional and metabolic changes both in the disease and in mAD. In addition, they added a short term experiment on mice, highlighting the differences between humans and mice. All this is an important and unprecedented piece of information that will certainly be useful to the scientific community in mitochondrial research.

I wish only to advise the authors that Figure 2B is not mentioned in the manuscript and I suggest them to check the text where they mention Figure 2A. I also recommend to check the method for the needle biopsy where the authors say that "The muscle samples for RNA and metabolomics analyses were snap-frozen in liquid nitrogen cooled isopentane for histology and stored at -80 {degree sign}C". I reckon part of the sentence is missing as usually muscle samples are snap frozen in liquid nitrogen, and samples for histochemistry in precooled isopentane. Moreover, the authors did not present data on muscle metabolomics and this should be removed.

Finally, I believe that the authors also correctly addressed the minor criticism from reviewer n.2, who was not available for this second revision. The experiment on mice extensively addresses the requested discussion on the differences between mice and humans, and the manuscript has been extensively revised addressing the grammatical/spelling mistakes.

2nd Revision - authors' response

10 August 2016

Referee #3 (Comments on Novelty/Model System):

The authors provide an unprecedented detailed description of the biochemical, morphological, transcriptional and metabolic changes both in the disease vs. the healthy controls and in mAD vs ND. In addition, they added a short term experiment on mice, highlighting the differences between humans and mice. All this is an important piece of information that will certainly have a profound impact in the field.

Au: Thank you, we appreciate this comment.

Referee #3 (Remarks):

In this new version, the authors improved their manuscript and addressed most of my concerns. It is really a pity that no biopsy is available at 2.5 years, but I understand this raised ethical concerns as not in the original informed consent. However, the authors provide detailed description of the biochemical, morphological, transcriptional and metabolic changes both in the disease and in mAD. In addition, they added a short term experiment on mice, highlighting the differences between humans and mice. All this is an important and unprecedented piece of information that will certainly be useful to the scientific community in mitochondrial research.

I wish only to advise the authors that Figure 2B is not mentioned in the manuscript and I suggest them to check the text where they mention Figure 2A.

Au: OK, revised.

I also recommend to check the method for the needle biopsy where the authors say that "The muscle samples for RNA and metabolomics analyses were snap-frozen in liquid nitrogen cooled isopentane for histology and stored at -80 C". I reckon part of the sentence is missing as usually muscle samples

are snap frozen in liquid nitrogen, and samples for histochemistry in precooled isopentane.

Au: OK, revised.

Moreover, the authors did not present data on muscle metabolomics and this should be removed.

Au: OK, revised – thank you for careful reading. Muscle metabolomics was accidentally mentioned in the methods, whereas serum metabolomics results are presented in the paper. Therefore, the serum data remain in the manuscript.

Finally, I believe that the authors also correctly addressed the minor criticism from reviewer n.2, who was not available for this second revision. The experiment on mice extensively addresses the requested discussion on the differences between mice and humans, and the manuscript has been extensively revised addressing the grammatical/spelling mistakes.

Au: Thank you, we agree.

Corresponding Author Name: Anu Suomalainen

Manuscript Number: EMM-2016-06592